# Effects of Land Use and Topographic Position on Soil Organic Carbon and Total Nitrogen Stocks in Different Agro-Ecosystems of the Upper Blue Nile Basin

**Getu Abebe [1,2,\*], Atsushi Tsunekawa [3] , Nigussie Haregeweyn [4] , Taniguchi Takeshi [3],
Menale Wondie [2] , Enyew Adgo [5] , Tsugiyuki Masunaga [6] , Mitsuru Tsubo [3],
Kindiye Ebabu [1,5], Mulatu Liyew Berihun [1,7] and Asaminew Tassew [5]**

[1]  The United Graduate School of Agricultural Sciences, Tottori University, 1390 Hamasaka, Tottori 680-8553, Japan; kebabu2@gmail.com (K.E.); mulatuliyew@yahoo.com (M.L.B.)

[2]  Amhara Agricultural Research Institute, Forestry Research Department, P.O. Box 527 Bahir Dar, Ethiopia; menalewondie@yahoo.com

[3]  Arid Land Research Center, Tottori University, 1390 Hamasaka, Tottori 680-0001, Japan; tsunekawa@tottori-u.ac.jp (A.T.); takeshi@alrc.tottori-u.ac.jp (T.T.); tsubo@tottori-u.ac.jp (M.T.)

[4]  International Platform for Dryland Research and Education, Tottori University, 1390 Hamasaka, Tottori 680-0001, Japan; nigussie_haregeweyn@yahoo.com

[5]  College of Agriculture and Environmental Sciences, Bahir Dar University, P.O. Box 1289 Bahir Dar, Ethiopia; enyewadgo@gmail.com (E.A.); atassew2005@yahoo.com (A.T.)

[6]  Faculty of Life and Environmental Science Shimane University, Shimane Matsue 690-0823, Japan; masunaga@life.shimane-u.ac.jp

[7]  Faculty of Civil and Water Resource Engineering, Bahir Dar Institute of Technology, Bahir Dar University, P.O. Box 26 Bahir Dar, Ethiopia

\*  Correspondence: gabebe233@gmail.com

**Abstract:** Soil organic carbon (SOC) and total nitrogen (TN) are key ecological indicators of soil quality in a given landscape. Their status, especially in drought-prone landscapes, is associated mainly with the land-use type and topographic position. This study aimed to clarify the effect of land use and topographic position on SOC and TN stocks to further clarify the ecological processes occurring in the landscape. To analyze the status of SOC and TN, we collected 352 composite soil samples from three depths in the uppermost soil (0–50 cm) in four major land-use types (bushland, cropland, grazing land, and plantation) and three topographic positions (upper, middle, and lower) at three sites: Dibatie (lowland), Aba Gerima (midland), and Guder (highland). Both SOC and TN stocks varied significantly across the land uses, topographic positions, and agro-ecosystems. SOC and TN stocks were significantly higher in bushland (166.22 Mg ha$^{-1}$) and grazing lands (13.11 Mg ha$^{-1}$) at Guder. The lowest SOC and TN stocks were observed in cropland (25.97 and 2.14 Mg ha$^{-1}$) at Aba Gerima, which was mainly attributed to frequent and unmanaged plowing and extensive biomass removal. Compared to other land uses, plantations exhibited lower SOC and TN stocks due to poor undergrowth and overexploitation for charcoal and firewood production. Each of the three sites showed distinct characteristics in both stocks, as indicated by variations in the C/N ratios (11–13 at Guder, 10–21 at Aba Gerima, and 15–18 at Dibatie). Overall, land use was shown to be an important factor influencing the SOC and TN stocks, both within and across agro-ecosystems, whereas the effect of topographic position was more pronounced across agro-ecosystems than within them. Specifically, Aba Gerima had lower SOC and TN stocks due to prolonged cultivation and unsustainable human activities, thus revealing the need for immediate land management interventions, particularly targeting croplands. In a heterogeneous environment such as the Upper Blue Nile basin, proper

understanding of the interactions between land use and topographic position and their effect on SOC and TN stock is needed to design proper soil management practices.

**Keywords:** *Acacia decurrens*; Eucalyptus; drought-prone; highland; midland; lowland

## 1. Introduction

Soil organic carbon (SOC) and total nitrogen (TN) provide information on the impact of land management on soil health. The SOC stock, which is a key component and the largest carbon pool in terrestrial ecosystems, is strongly linked to nitrogen availability [1] and serves as an indicator of soil quality [2]. SOC acts as a major source or sink for atmospheric CO2 [3–6]. Globally, soil is estimated to store 3150 Pg C (1 Pg C = 1015 g C), which is four times greater than carbon storage in the terrestrial plant biomass (650 Pg C) and atmospheric (750 Pg C) pools [7]. The size of the soil carbon pool, however, is significantly controlled by the balance between the input and output of carbon in an ecosystem. Therefore, any change in the size of the SOC stock potentially affects elemental cycling, land productivity, atmospheric CO2 concentration, and thus global climate [7–9].

The amount of SOC in a terrestrial ecosystem is influenced by natural and anthropogenic factors [10]. Human-induced land-use change causes a particularly substantial loss of SOC [7,11,12]. Land-use change is associated with ecosystem carbon change [13] and drives negative impacts on climate and the environment. Numerous studies have shown that deforestation and land-use change results in land degradation and poorer soil quality [8,14–16]. In Ethiopia, the conversion of natural vegetation to croplands or plantations is increasing due to population pressure and socio-economic drivers. This has implications for biodiversity decline, land productivity, desertification, and SOC dynamics [5,17–20]. According to Assefa et al. [5], conversion of natural forest to cropland in the northern highland of Ethiopia accounted for 50% to 87% of the observed SOC reduction. Likewise, Kassa et al. [21] reported that conversion of forest and agroforestry to croplands caused an annual decline of SOC stock from 3.3 to 8.0 Mg ha$^{-1}$ in the southwestern highlands of Ethiopia. On the other hand, reports on vegetation restoration of degraded lands in the region indicated that SOC is improved by planting Eucalyptus trees [18,22,23] or establishing exclosures [24,25].

Generally, the soil of natural vegetation has higher SOC than croplands because of its higher organic residue content [26]. However, the efficiency of SOC accumulation depends on the quality and amount of organic inputs, decomposition rate in the soil [26], and topographic position [10,27]. Topography influences SOC mainly by altering the input and output of carbon via hydrological processes, and it affects soil erosion and sediment deposition [28]. The topographic position also affects water availability, temperature regime, vegetation distribution, and soil processes [15,29].

Recently, owing to their strong influence on the sustainability of natural and agricultural ecosystems, the effects of factors such as land use, topography, and their interaction on SOC and TN stocks have attracted scientific attention at the small watershed scale [10,12,13,15,27,30]. At the regional scale, climate is the dominant factor that controls SOC and TN stocks by inducing changes in soil moisture, vegetation patterns, decomposition rate [31], microbial activity [32], and soil respiration [33]. Therefore, SOC and TN dynamics in the soil vary in response to environmental factors (both biotic and abiotic), and are sensitive to changes in climate and the local environment [1]. Thus, understanding soil carbon and nitrogen stock dynamics in different agro-ecosystems as a function of topographic position, land use, and their interaction is important for designing sustainable land management options [22,27,34] that also contribute to food security [35].

The direct and interactive effects of topography and land use on SOC and TN stocks are not well studied in the landscape of Ethiopia's Upper Blue Nile basin, which is also known as the Abay River basin and covers an area of 173,000 km$^2$ [36]. The climate of the region is tropical highland monsoonal [37]. The region is characterized by fragile and drought-prone areas, with diverse

agro-ecosystems and severe land degradation. Although soil and water conservation practices have been used since the 1980s [38], a reduction in vegetation cover [39,40] and soil erosion induced by poor land-use management have become major challenges for ensuring food security [36].

The aim of this study was to assess the effects of major controlling factors on SOC and TN stocks in three agro-ecosystems of the Upper Blue Nile basin. The specific objectives were to (1) determine how stocks of SOC and TN vary with topographic position, land-use type, and soil depth across agro-ecosystems; (2) assess the interactive effect of land use and topographic position on SOC and TN stocks within and across agro-ecosystems; and (3) assess the current spatial distribution of SOC and TN stocks in the three agro-ecosystems of the Upper Blue Nile basin.

## 2. Materials and Methods

### 2.1. Study Sites

The study was conducted in three different agro-ecosystems of the Upper Blue Nile basin, Ethiopia (Figure 1), namely Guder, Aba Gerima, and Dibatie, representing the highland, midland, and lowland agro-ecosystems, respectively (Table 1). According to Mekonnen [41], the four dominant soil types (in the FAO classification system) in the study area are Acrisols, Leptosols, Luvisols, and Vertisols (Table 1).

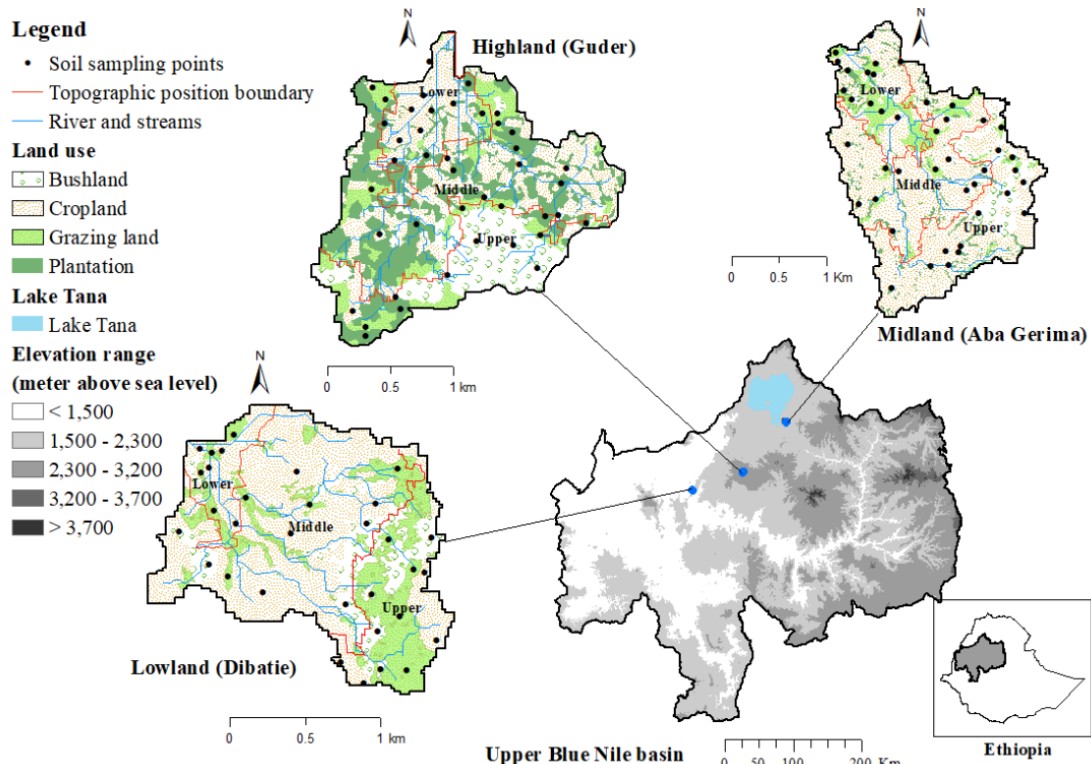

**Figure 1.** Location of the three study sites in the Upper Blue Nile basin, with respective land-use and drainage maps shown. The points in each watershed illustrate the distribution of sampling points with respect to land use and three topographic positions.

In the Koppen–Geiger classification [42], the climate is characterized as subtropical oceanic highland at Guder, humid subtropical at Aba Gerima, and tropical wet-dry at Dibatie. The rainfall pattern is unimodal and mostly occurs from June to September at all sites (Figure 2). The mean annual rainfall was 1022, 1343, and 2454 mm at Dibatie, Aba Gerima, and Guder, respectively. Mean annual temperature varies from 25 to 32 °C at Dibatie, from 13 to 27 °C at Aba Gerima, and from 9.4 to 25 °C at Guder [43,44].

**Table 1.** Site characteristics of study watersheds in the Upper Blue Nile basin.

| Site Characteristics | Site (Watershed) | | |
|---|---|---|---|
| | Guder (Highland) | Aba Gerima (Midland) | Dibatie (Lowland) |
| Longitude, latitude | 11°0′35.13″ N, 36°56′7.97″ E | 10°45′53.09″ N, 36°16′19.11″ E | 11°39′27.26″ N, 37°30′14.21″ E |
| Area (ha) | 343 | 426 | 246 |
| Elevation (m a.s.l.) | 2500–2800 | 1900–2200 | 1400–1700 |
| Slope gradients (°) [a] | 0–32 | 0–36 | 0–21 |
| Topographic positions (elevation range and mode of slope (%)) | | | |
| Upper | (2500–2600, 30–50) | (2200–2100, 10–20) | (1700–1600, 10–20) |
| Middle | (2600–2700, 10–20) | (2100–2000, 10–20) | (2100–2000, 0–10) |
| Lower | (2700–2800, 0–10) | (2000–1900, 0–10) | (2000–1900, 0–10) |
| Annual mean temperature (°C) [b] | 9.4–25 | 13–27 | 25–32 |
| Rainfall (mm yr$^{-1}$) [b] | 1951–3424 | 895–2037 | 850–1200 |
| Agro-ecology [c] | oceanic subtropical | humid subtropical | tropical wet-dry |
| Soil parent material [d] | Basalt (Quaternary) | Basalt (Oilgo pilocene) | - |
| Major soil types [e] | Acrisols and Leptosols | Leptosols and Luvisols | Luvisols and Vertisols |
| Primary soil texture [a] | clay loam | clay | clay |
| Sand, silt, and clay (%) [e] | 30, 40, and 30 | 15, 30, and 55 | 25, 19, and 56 |
| Selected soil properties | | | |
| pH (water) | 4.2–6.5 | 4.7–6.8 | 5.8–7.4 |
| Electric conductivity (dS m$^{-1}$) | 0.01–0.11 | 0.01–0.12 | 0.02–0.19 |
| Cation exchange capacity (cmol kg$^{-1}$) [e] | 21.4–65.7 | 23.8–26.8 | 23.2–48.8 |
| Land-use types (Area, ha) | bushland (58.8), cropland(106), grazing land (47.1), plantation forest (116.5) | bushland (46.5), cropland (220), grazing land (14.6), plantation forest (38) | bushland (37.6, cropland (151), grazing land (55.3) |
| Dry biomass (tones ha$^{-1}$ yr$^{-1}$) [f] | | | |
| Cropland (teff) | 7.14 | 6.17 | 6.94 |
| Grazing land | 3.9 | 3.08 | 7.9 |

[a] Slope and soil texture data taken from [45]. [b] Weather data (1999–2015) was obtained from [43]. [c] Koppen–Geiger classification [42]. [d] Soil geology data taken from [46]; [e] Soil characteristics taken from [41]. [f] Dry biomass data was obtained from KAKENHI project (average from 150 plots (1 m × 3 m), 2016–2017).

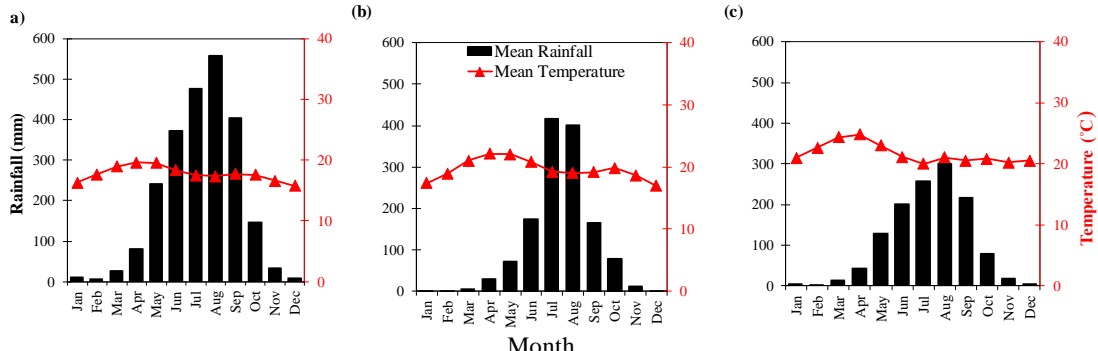

**Figure 2.** Climograph of Guder (**a**), Aba Gerima (**b**), and Dibatie (**c**) from 1999 to 2017.

The native tree and shrub species common at Guder are *Acacia abyssinica*, *Albizia gummifera*, *Croton macrostaches*, *Combretum molle*, *Cordia africana*, *Schefflera abyssinica*, *Dovyalis abyssinica*, and *Entada abyssinica*. Those at Aba Gerima are *A. gummifera*, *Bersama abyssinica*, *Calpurnia aurea*, *Croton macrostaches*, *Olea europaea*, *Ficus thonningii*, and *E. abyssinica*. At Dibatie, *Acacia negrii*, *Acacia sieberiana*, *Ficus sycomorus*, *Terminalia brownii*, *Terminalia schimperiana*, and *Oxytenanthra abyssinica* are common in the bushlands. *A. decurrens* at Guder and *Eucalyptus camadulensis* at Aba Gerima are the dominant exotic tree species planted as woodlots for fuelwood, charcoal, and construction wood production. Clear felling (Guder) and coppice management (Aba Gerima) are the common plantation management practices. The rotation period of the plantations at Guder and Aba Gerima is 3–5 and 5–7 years, respectively.

Rainfed, subsistence-based and mixed farming (crop cultivation and livestock rearing) is the main agricultural practice at the study sites [44]. At Guder, teff (*Eragrostis tef*), barley (*Hordeum vulgare*),

wheat (*Triticum aestivum*), and potato (*Solanum tuberosum*) are grown. At Aba Gerima, teff, finger millet (*Eleusine coracana*), wheat, and maize (*Zea mays*) are cultivated. At Dibatie, maize, teff, sorghum (*Sorghum bicolor*), and groundnut (*Arachis hypogaea*) are the major food crops [43,45].

## 2.2. Soil Sampling

Based on the available land-use types and elevation range of the watersheds (Table 1), three topographic positions (i.e., upper, middle, and lower) were selected. Cropland and grazing land are common in all topographic positions at the three study sites, whereas plantation (*A. decurrens* or *E. camaldulensis*) at Dibatie and bushland in the lower position at all sites are not part of the current land-use systems (Figure 1). In each topographic position, four replicated land uses were measured. A total of 352 soil samples were collected from the three agro-ecosystems. The top 50 cm of soil was sampled, divided into three soil layers of 0–15, 15–30, and 30–50 cm. Soil samples were collected from five points, at the four corners and in the center of a plot (10 m × 10 m) using a hand-driven soil auger. Soil samples collected from each plot from similar layers were thoroughly mixed to obtain a composite sample (1 kg). Soil bulk density was determined separately by using a metal core cylinder (100 cm$^3$), which was inserted at the midpoint of the 0–15, 15–30, and 30–50 cm layers. All composite soil samples were first air-dried and then passed through a 2-mm sieve, packed, labeled, and transported to Japan for chemical analysis at the Arid Land Research Center of Tottori University.

## 2.3. Soil Analysis

Soil pH and electrical conductivity were measured at a 1:5 soil-to-water ratio using a pH meter (D-51, Horiba, Kyoto, Japan) and conductivity meter (ES-51, Horiba), respectively. Bulk soil density (Mg m$^{-3}$) was determined for core soil samples after oven-drying at 105 °C for 24 h.

## 2.4. Determination of SOC and TN Stocks

Five-gram subsamples of homogenized soil from each soil depth were dried at 60 °C for 48 h. From each subsample, 1 g of soil was taken, and total organic carbon and nitrogen were determined using a CN corder (Macro Corder JM1000CN, J-Science Lab, Kyoto Japan). Total carbon and nitrogen stocks (Mg ha$^{-1}$) down to the 50 cm soil horizon were calculated using the model of [47]:

$$\text{SOC (or TN) stock} = \text{content} \times \rho b \times d \times 10{,}000 \text{ m}^2 \text{ ha}^{-1} \times 0.001 \text{ Mg kg}^{-1}, \tag{1}$$

where SOC (or TN) stock is the soil organic carbon or total nitrogen stock (Mg ha$^{-1}$), content is the soil organic carbon or total nitrogen concentration (kg Mg$^{-1}$), $\rho b$ is the soil bulk density (Mg m$^{-3}$), and d is the thickness of the soil layer (m).

## 2.5. Data Analysis

Data with a non-normal distribution were transformed using square-root and log transformation techniques. Two-way (within agro-ecosystem) and nested three-way (between agro-ecosystems) analysis of variance were used to test the significance of mean differences in SOC and TN content and stock as dependent variables, while topographic position, land use, soil depth, and their interactions (between two or three factors) were considered as driving factors. Differences in means between groups were analyzed using Tukey's HSD (honestly significant difference) test within the Agricolae package (version 1.2-8). Statistical analyses were carried out in RStudio [48], an interface for the R software program (version 3.4.4). The significance level was set at alpha = 0.05.

## 3. Results

### 3.1. Effect of Topographic Position on SOC and TN Contents and Stocks

At Guder, SOC content in croplands increased significantly ($p < 0.05$) from the upper (10.96 mg g$^{-1}$) to the lower topographic position (16.68 mg g$^{-1}$; Figure 3a). In the case of grazing land, SOC content decreased from 22.59 mg g$^{-1}$ in the upper position to 14.57 mg g$^{-1}$ in the middle position and then increased to 17.40 mg g$^{-1}$ in the lower position. For bushland and *A. decurrens* plantations, SOC content did not vary among topographic positions. TN content for bushland decreased significantly from 2.87 mg g$^{-1}$ in the upper position to 2.39 mg g$^{-1}$ in the middle position (Figure 3d). However, TN in cropland and grazing lands were not significantly different across topographic position.

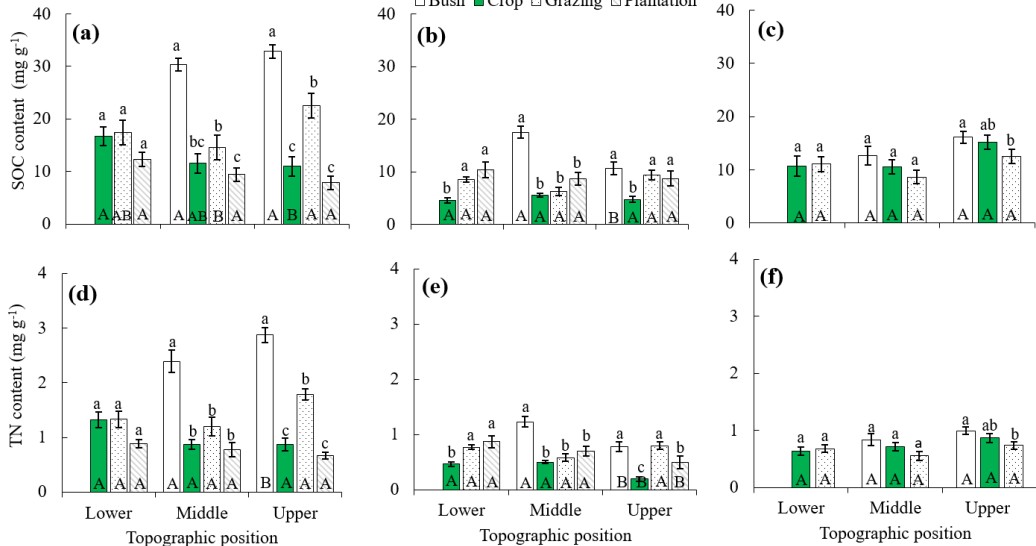

**Figure 3.** Soil organic carbon (SOC) and total nitrogen (TN) contents under different land uses and topographic positions at Guder (**a**,**d**), Aba Gerima (**b**,**e**), and Dibatie (**c**,**f**). Different lowercase letters above the bars indicate significant differences in SOC and TN among land uses in the same topographic position ($p < 0.05$); different capital letters indicate significant differences in SOC and TN among topographic positions within the same land use ($p < 0.05$). Error bars represent the standard error of the mean at alpha = 0.05.

At Aba Gerima, SOC and TN contents differed among topographic positions ($p < 0.05$; Figure 3b,e). The TN contents of bushland (0.78 mg g$^{-1}$), cropland (0.19 mg g$^{-1}$), and Eucalyptus plantations (0.49 mg g$^{-1}$) were significantly lower at the upper position than that at the middle and lower positions. The highest contents of SOC (17.52 mg g$^{-1}$) and TN (1.23 mg g$^{-1}$) were in bushland at the middle position, whereas croplands in the upper position showed the lowest SOC (4.78 mg g$^{-1}$) and TN contents (0.19 mg g$^{-1}$).

At Dibatie, SOC and TN contents in the upper position in bushland and cropland were 16.12 and 0.99, and 15.22 and 0.86 mg g$^{-1}$ higher, respectively, than those in grazing land (12.58 and 0.74 mg g$^{-1}$; Figure 3c,f). In contrast, both SOC and TN contents were similar among land-use types in the middle and lower topographic positions.

At Guder, the SOC stock decreased from the upper to lower topographic positions in grazing land (Figure 4a). SOC stock under *A. decurrens* plantations increased significantly ($p < 0.05$), from 42.73 Mg ha$^{-1}$ in the upper position to 44.63 Mg ha$^{-1}$ in the middle position and 58.94 Mg ha$^{-1}$ in the lower position. The SOC stock in bushland was highest (166.22 Mg ha$^{-1}$) in the upper position. The TN stock was significantly higher (13.11 Mg ha$^{-1}$) in grazing lands in the upper position and decreased toward the lower position (Figure 4d).

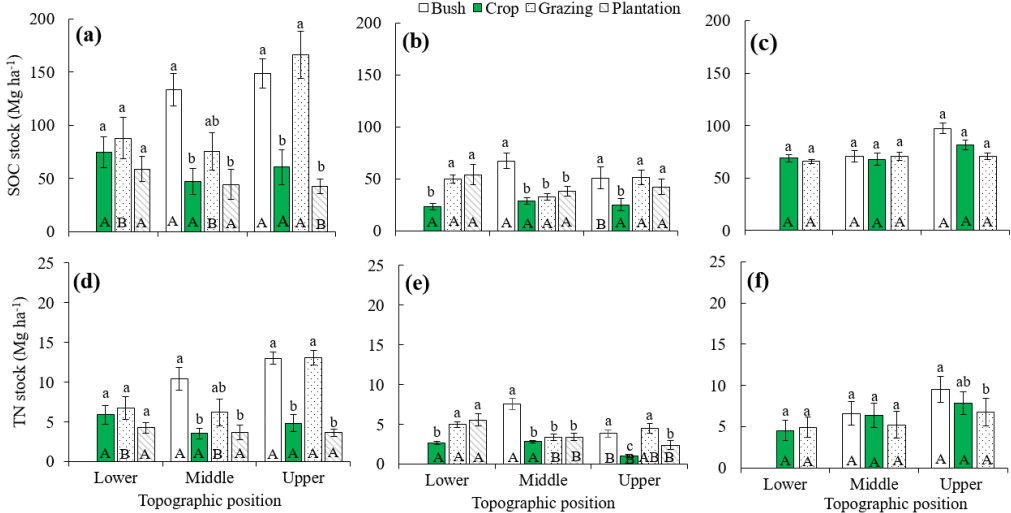

**Figure 4.** Soil organic carbon (SOC) and total nitrogen (TN) stocks under different land-use types and topographic positions at Guder (**a**,**d**), Aba Gerima (**b**,**e**), and Dibatie (**c**,**f**). Different lowercase letters above the bars indicate significant differences in SOC and TN among land-use types in the same topographic position (*p < 0.05*); different capital letters indicate significant differences in SOC and TN among topographic positions within the same land-use type ($p < 0.05$). Error bars represent the standard error of the mean at alpha = 0.05.

At Aba Gerima, TN stocks in cropland and Eucalyptus plantations increased significantly from the upper to lower positions (Figure 4e).

At Dibatie, TN stocks in bushland, cropland, and grazing lands varied significantly between the upper and the middle and lower positions (Figure 4f). The highest (9.53 Mg ha$^{-1}$) and lowest (4.56 Mg ha$^{-1}$) TN stocks were recorded in bushlands and croplands in the upper and middle positions, respectively.

In general, SOC and TN stocks at Guder, TN content and stock at Aba Gerima, and SOC and TN contents at Dibatie were influenced by topographic position ($p < 0.05$; Table 2). With the exception of SOC stock at Dibatie, both SOC and TN contents and stocks at the three study sites were strongly affected by land use ($p < 0.05$). Likewise, the interaction between topographic position and land use had a significant effect on both SOC and TN contents and stocks at Guder and Aba Gerima, whereas no significant effect was detected at Dibatie (Table 2).

**Table 2.** Results of two-way analysis of variance (ANOVA) for SOC and TN contents and stocks as a function of topographic position and land use in different agro-ecosystems of the Upper Blue Nile basin.

| Agro-Ecosystem | Source | df | *p*-Value | | | |
| --- | --- | --- | --- | --- | --- | --- |
| | | | SOC Content | TN Content | SOC Stock | TN Stock |
| | topographic position | 2 | 0.278 | 0.093 | <0.001 | <0.001 |
| Guder | land use | 3 | <0.001 | <0.001 | <0.001 | <0.001 |
| | topographic position × land use | 5 | 0.034 | 0.019 | <0.001 | <0.001 |
| | topographic position | 2 | 0.562 | <0.001 | 0.307 | <0.001 |
| Aba Gerima | land use | 3 | <0.001 | <0.001 | <0.001 | <0.001 |
| | topographic position × land use | 5 | 0.012 | 0.002 | <0.001 | <0.001 |
| | topographic position | 2 | 0.010 | 0.003 | 0.046 | 0.005 |
| Dibatie | land use | 2 | 0.024 | 0.003 | 0.492 | 0.004 |
| | topographic position × land use | 3 | 0.924 | 0.681 | 0.766 | 0.793 |

Notes: Topographic positions: upper, middle, and lower; land uses: bushland, cropland, grazing land, and plantation.

### 3.2. SOC and TN Contents and Stocks for Different Land Uses across Soil Depths

Across soil profiles, both SOC and TN contents were slightly decreased from top to the lower soil profile at Aba Gerima (Figure 5b,e) compared with Guder (Figure 5a,d) and Dibatie (Figure 5c,f).

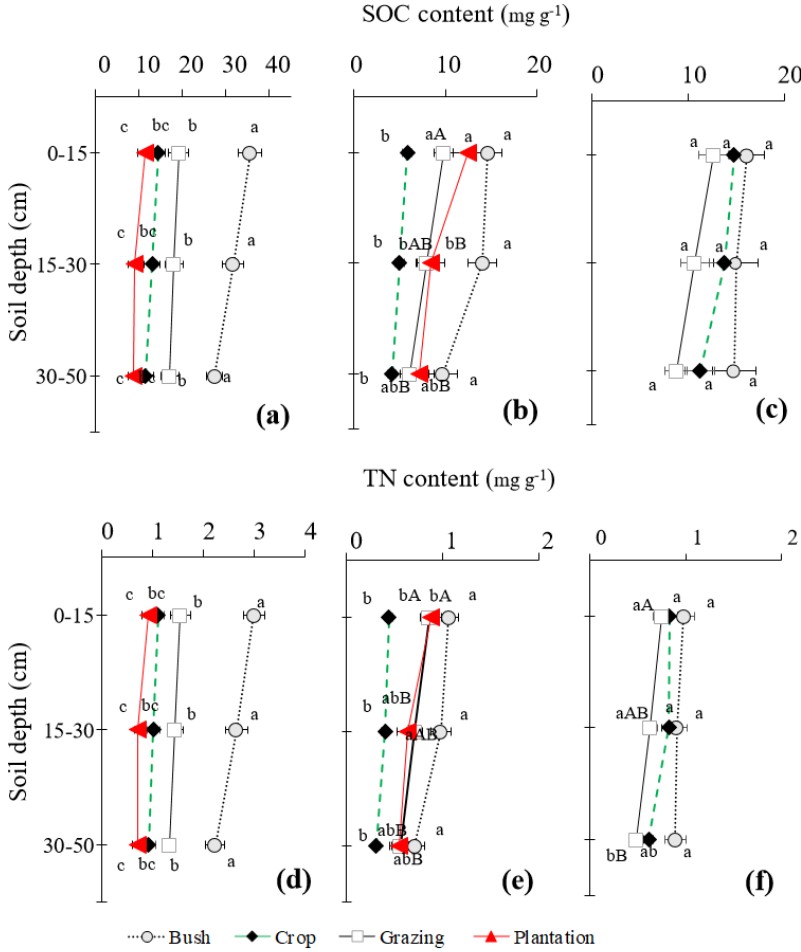

**Figure 5.** Soil organic carbon and total nitrogen contents in relation to land-use type at Guder (**a**,**d**), Aba Gerima (**b**,**e**), and Dibatie (**c**,**f**). Different lowercase letters indicate significant differences in SOC; n = 12).

The SOC and TN contents at Guder (Figure 5a,d) and Aba Gerima (Figure 5b,e) varied significantly among land uses at all soil depths ($p < 0.05$). At Dibatie, except for TN contents in the lower soil depth, there were no significant differences in SOC and TN contents among land-use types at all soil depths (Figure 5c,f).

At Guder, SOC and TN contents of bushland were significantly higher than the other land uses in all soil profiles (Figure 5a,d). At Aba Gerima, SOC contents in the 0–15 cm layer were 15, 11.3, and 6.7 times higher in bushland, plantation, and grazing land, respectively, than in cropland (Figure 5b). However, in the 15–30 and 30–50 cm layers, the SOC content in bushland was significantly greater than that in the other land uses. Similarly, TN content in the 0–15 cm layer at Aba Gerima was significantly higher ($p < 0.05$; 1.06 mg g$^{-1}$) in bushland than in the other land uses (Figure 5e). In the 15–30 and 30–50 cm soil layers, the TN content was 1.29, 0.81, and 0.77 times higher in bushland, grazing land, and plantation, respectively, than in cropland ($p < 0.05$). At Dibatie, there were no significant differences in SOC content among land-use types at all soil depths (Figure 5c). However, TN contents in the 30–50 cm soil layer were significantly higher in bushland and cropland than in grazing land (0.49 mg g$^{-1}$; $p < 0.05$; Figure 5f).

On the other hand, the SOC and TN stocks at Guder and Aba Gerima varied significantly among land uses within each soil profile ($p < 0.05$; Table S1). Significant differences in SOC stocks across soil depths within each land-use type were observed at Aba Gerima and Dibatie, whereas TN stock only varied significantly at Dibatie (Table S1). SOC stocks of grazing land and plantations showed a 0.43- and 0.44-fold decrease from the top layer to the lower layer, respectively, at Aba Gerima. The SOC stock in cropland decreased significantly across soil depths at Dibatie. TN stock in the 0–15 cm layer was higher than at 15–30 and 30–50 cm soil depths at Dibatie, whereas no significant difference was found in the TN stock between the 15–30 and 30–50 cm layers.

### 3.3. Effect of Agro-ecosystem on SOC and TN Contents and Stocks

SOC and TN contents and stocks in bushland and grazing land at Guder were significantly larger than those at Dibatie and Aba Gerima (Table 3). SOC content in croplands at Aba Gerima (5.01 mg g$^{-1}$) was significantly lower than that at Guder (13.07 mg g$^{-1}$) and Dibatie (13.28 mg g$^{-1}$). Plantation at Guder (*A. decurrens*) and Aba Gerima (Eucalyptus) had similar SOC and TN stocks (Table 3). In contrast, the TN stock in cropland was significantly higher at Dibatie (6.26 Mg ha$^{-1}$) than at Guder (4.77 Mg ha$^{-1}$) and Aba Gerima (2.14 Mg ha$^{-1}$).

Both SOC and TN contents in the lower and middle topographic positions were significantly higher at Guder than at Aba Gerima and Dibatie (Table S2), whereas the SOC and TN contents in the upper position were significantly lower at Aba Gerima than at Guder and Dibatie. The SOC stocks in the upper position were in the following order: Guder (104.67 Mg ha$^{-1}$), Dibatie (88.90 Mg ha$^{-1}$), Aba Gerima (41.91 Mg ha$^{-1}$). The SOC stock in the middle and lower positions and TN stock in all topographic positions were significantly lower at Aba Gerima, but the values for Guder and Dibatie were similar (Table S2).

There was significant variation in the C/N ratios of cropland, grazing land, and plantation among sites (Table 3). The highest (17.52) and lowest (11.03) C/N ratios were those of grazing lands at Dibatie and Aba Gerima, respectively. Eucalyptus plantations (12.15) showed a significantly higher C/N ratio than *A. decurrens* plantations (12.16). The C/N ratios in the upper position were significantly higher at Aba Gerima followed by Dibatie and Guder, whereas the C/N ratios in the middle and lower positions were significantly higher at Dibatie than at the other two sites (Table S2).

Bulk densities in bushland, cropland, and grazing land differed significantly among sites ($p < 0.05$; Table 3), whereas those of plantations at Guder (*A. decurrens*) and Aba Gerima (*Eucalyptus*) were not significantly different. Bulk density ranged from 0.90 to 1.18 Mg m$^{-3}$, from 1.09 to 1.19 Mg m$^{-3}$, and from 1.11 to 1.32 Mg m$^{-3}$ at Guder, Aba Gerima, and Dibatie, respectively (Table 3). Bulk densities in bushland, cropland, and grazing land were significantly lower at Guder than at Aba Gerima and Dibatie (Table 3). The bulk density also varied significantly among topographic positions ($p < 0.05$; Table S2). Soils at Guder showed significantly lower soil bulk density in the lower and middle positions as compared to the upper position. Soils at Dibatie had significantly higher soil bulk density in the middle position (Table S2).

Overall, SOC and TN contents and stocks were strongly dependent on agro-ecosystem ($p < 0.05$), land use ($p < 0.05$), and the interaction between agro-ecosystem and land use ($p < 0.05$; Table 4). Topographic position ($p < 0.05$) also influenced SOC content, SOC stock, and TN stock, but not TN content. In addition, the interaction between agro-ecosystem and topographic position affected TN content and SOC and TN stocks. TN content, SOC stock, and TN stock were also strongly dependent on the interaction of agro-ecosystem, topographic position, and land use.

**Table 3.** SOC and TN contents and stocks in different land-use types at the three study sites.

| Land Use | Site | SOC | | TN | | C/N Ratio | Bulk Density |
|---|---|---|---|---|---|---|---|
| | | mg g$^{-1}$ | Mg ha$^{-1}$ | mg g$^{-1}$ | Mg ha$^{-1}$ | | Mg m$^{-3}$ |
| Bushland | Guder | 31.63 (1.45) [a] | 141.19 (6.74) [a] | 2.63 (0.13) [a] | 11.73 (0.50) [a] | 12.16 (0.30) [a] | 0.90 (0.01) [b] |
| | Aba Gerima | 13.42 (1.16) [b] | 59.23 (7.19) [b] | 0.96 (0.09) [b] | 4.13 (1.91) [b] | 14.01 (0.31) [a] | 1.09 (0.03) [a] |
| | Dibatie | 15.31 (1.19) [b] | 85.70 (9.81) [a,b] | 0.92 (0.06) [b] | 8.08 (1.51) [a] | 16.16 (0.57) [a] | 1.11 (0.04) [a] |
| Cropland | Guder | 13.07 (1.04) [a] | 61.00 (2.33) [a] | 1.02 (0.07) [a] | 4.77 (0.28) [b] | 12.52 (0.32) [b] | 0.96 (0.03) [b] |
| | Aba Gerima | 5.01 (0.29) [b] | 25.97 (7.04) [b] | 0.39 (0.03) [b] | 2.14 (1.44) [c] | 17.63 (2.33) [a] | 1.12 (0.01) [a] |
| | Dibatie | 13.28 (0.96) [a] | 72.62 (8.33) [a] | 0.75 (0.04) [a] | 6.26 (1.34) [a] | 17.38 (0.82) [a] | 1.12 (0.03) [a] |
| Grazing | Guder | 18.19 (1.20) [a] | 109.94 (3.69) [a] | 1.44 (0.09) [a] | 8.68 (0.31) [a] | 12.68 (0.20) [b] | 1.18 (0.04) [b] |
| | Aba Gerima | 7.97 (0.49) [b] | 44.14 (7.55) [b] | 0.71 (0.04) [b] | 4.17 (1.19) [b] | 11.03 (0.17) [b] | 1.19 (0.03) [b] |
| | Dibatie | 10.84 (0.81) [b] | 66.42 (4.69) [b] | 0.63 (0.05) [b] | 5.65 (1.52) [a,b] | 17.52 (0.72) [a] | 1.32 (0.02) [a] |
| Plantation | Guder | 9.85 (0.87) [a] | 48.79 (7.15) [a] | 0.78 (0.06) [a] | 3.86 (0.65) [a] | 12.16 (0.41) [b] | 1.02 (0.02) [a] |
| | Aba Gerima | 9.24 (0.79) [a] | 44.77 (6.33) [a] | 0.68 (0.06) [a] | 3.56 (0.38) [a] | 17.15 (2.62) [a] | 1.06 (0.03) [a] |

Mean (standard error) values were calculated across the whole soil depth from 0 to 50 cm. Within a column, different letters for each land use indicate a significant difference between sites (Tukey's HSD at $p < 0.05$).

**Table 4.** Nested three-way ANOVA for SOC and TN contents and stocks as functions of topographic position and land use across agro-ecosystems.

| Source | df | *p*-Value | | | |
|---|---|---|---|---|---|
| | | SOC Content | TN Content | SOC Stock | TN Stock |
| Agro-ecosystem | 2 | <0.001 | <0.001 | <0.001 | <0.001 |
| Topographic position | 2 | 0.043 | 0.862 | 0.019 | <0.001 |
| Land use | 3 | <0.001 | <0.001 | <0.001 | <0.001 |
| Agro-ecosystem × topographic position | 4 | 0.082 | <0.001 | 0.003 | <0.001 |
| Agro-ecosystem × land use | 5 | <0.001 | <0.001 | <0.001 | <0.001 |
| Agro-ecosystem × topographic position × land use | 13 | 0.058 | <0.001 | <0.001 | <0.001 |

Agro-ecosystems: Guder, Aba Gerima, and Dibatie; topographic positions: upper, middle, and lower; land uses: bushland, cropland, grazing land, and plantation.

## 4. Discussion

### 4.1. Effects of Land-Use Type on SOC and TN Contents and Stocks Across Topographic Positions

Climate, soil type, land use, and topography are the principal factors that control SOC and TN distributions at a regional scale [49,50]. In a small watershed, however, soil type and climate variability are commonly low [51]. Our findings confirmed that land use and topography influenced the SOC and TN storage in the three agro-ecosystems.

At Guder, the SOC content in cropland was significantly increased (Figure 3a) from the upper to lower topographic positions. In fact, the upper position of a watershed is often exposed to soil erosion, serving as a source of run-off and sediment for the lower positions [10]. In cropland, particularly, this situation has amplified the variation of SOC content in association with geomorphologic processes. Cropland in the highlands is poor in vegetation cover and experiences soil disturbance due to tillage and high biomass removal [5,22]. In contrast, SOC content in grazing land was significantly decreased from the upper to lower positions (Figure 4a). Less soil disturbance, greater vegetation cover, and organic input from grazing animals would improve the SOC in the upper position. Similarly, Mekuria et al. [25] reported better vegetation cover and biomass in communal grazing lands in the upper position than in the lower position, which is more easily accessed by livestock that induce changes in SOC. Zhu et al. [30] also found an increasing trend in SOC content for cropland and a decreasing trend in grassland from the summit to the lower part of a watershed in China. A review by Deng et al. [52] of studies conducted worldwide revealed that conversion of native vegetation to grassland significantly increased the SOC stock. In contrast, the SOC content in bushland and plantation at Guder were not affected by topographic position. This distribution pattern may be due to the generally good vegetation cover in bushland and plantations, which may reduce soil erosion in the upper position, resulting in similar SOC contents in the middle and lower positions. Likewise, Fu et al. [13] reported uniform SOC contents under different vegetative types along a hillslope on the Loess Plateau of China.

The TN content in bushland at Guder was higher in the upper than middle position, which was probably due to the presence of a large number of native leguminous shrubs (e.g., *E. abyssinica*) and trees (e.g., *A. abyssinica* and *A. gummifera*) in the bushland. These results correspond with the finding of [21], who reported high TN content in native vegetation consisting of leguminous tree species. TN stock of the grazing land was significantly higher (13.11 Mg ha$^{-1}$) in the upper position than in the middle and lower positions (Figure 4d), likely because grazing land in the upper position was recently converted from bushland [40], which may have stored relatively high SOC and TN stocks.

At Aba Gerima, both SOC and TN contents were significantly different among land uses (Figure 3b,e). SOC and TN contents in the middle position of bushland (17.52 and 1.23 mg g$^{-1}$, respectively) were higher than those in the upper position. Similarly, a study conducted in northern Ethiopia [53] reported higher SOC and TN contents in the middle position of natural vegetation. Our results could be associated with soil erosion, which is a common problem in the study area and elsewhere in Ethiopia [36,43]. Soil erosion often causes translocation of soil from the upper slope

to lower area and contributes to the loss of soil organic matter [15]. Many studies elsewhere in the world [30,53,54] have reported that soil in sites of deposition has higher SOC and TN stocks.

At Dibatie, both SOC and TN contents in the upper position were significantly affected by land-use type (Figure 3c,f), which could be due to greater anthropogenic pressures in the upper than in the middle and lower topographic positions. Many members of the farming community live around the lower part of the watershed and their livelihoods depend on the bushland. This results in continuous removal of wood and bushland clearing for cropland and grazing land toward the upper position. SOC contents in bushland and cropland were 16.12 and 15.22 mg g$^{-1}$ higher and TN contents were 0.99 and 0.86 mg g$^{-1}$ higher than those of grazing lands. Natural vegetation at Dibatie is dominated by deciduous tree and shrub species that commonly contribute large amounts of organic matter to the soil. However, grassland is regularly burned, which substantially reduces the grass cover and induces loss of SOC and TN contents [55]. In a study in Ethiopia, [56] reported that the natural vegetation in Dibatie (Combretum–Terminalia) decreased as a result of fire.

With regard to land-use effects, the SOC stock of grazing land soil decreased significantly from the upper (162.22 Mg ha$^{-1}$) to middle positions (75.50 Mg ha$^{-1}$) at Guder. Soil bulk density in grazing lands is relatively higher as a result of livestock trampling [22,24,57]. At this site, a bulk density of 1.18 Mg/m$^3$ was recorded in the grazing land (Table 3). At Aba Gerima, a high SOC stock was stored in the middle position of bushland. Similarly, at Dibatie bushlands showed higher SOC and TN contents than those of the other land-use types.

## 4.2. Effect of Soil Depth on SOC and TN Contents and Stocks

At Guder, SOC and TN contents at 0–15 cm soil depth in bushland were higher than those of other land-use types (Figure 5a,d). Bushland comprises a sizable proportion of native vegetation, and the bushes, shrubs, and trees contain a substantial amount of wood biomass with a lower decomposition rate, which could improve the organic input and contribute more to soil SOC and TN. These results are similar to those of previous studies [10,21] that reported higher SOC and TN contents in the surface soil under native vegetation as compared to that of other land uses. Therefore, conversion of bushland to another land-use type may cause a substantial amount of SOC loss from the surface soil, as reported by studies conducted elsewhere [3,4,21].

Plantation (*A. decurrens* woodlot) contained lower surface SOC and TN contents than we expected (Figure 5a,d). Tesfaye et al. [58] reported lower SOC and TN contents in *A. decurrens* plantations in the central highlands of Ethiopia, which reflects the complete removal of plant residues from the woodlots. The plantations were established on previous cropland areas, but due to prolonged soil disturbance and soil erosion, this land was no longer able to support crop production. Thus, farmers had to change the cropland to plantations as a result of poor soil fertility and degradation [59,60]. Because plantations are commonly used for charcoal production, both the above- and belowground biomass is completely removed at the end of a rotation cycle (~3–5 years). According to Sultan et al. [61], plantations have high stand density (<1 m spacing), no understory vegetation cover, poor infiltration, and high runoff, all of which could contribute to their lower SOC and TN contents.

Similarly, in the 0–15 cm layer of Aba Gerima, cropland has significantly less surface SOC and TN contents than bushland, plantation, and grazing land (Figure 5b,e). This difference may be due to croplands having less organic input than areas with more vegetation. However, plantations at Aba Gerima had SOC contents comparable to those of cropland and grazing lands. These differences in SOC content from the plantations at Guder are likely induced by the differences in species and woodlot management. Unlike the Acacia plantations at Guder, the plantations at Aba Gerima consist of *Eucalyptus camaldulensis*, and tree harvesting operations do not include the belowground biomass. A study in northern Ethiopia revealed that Eucalyptus plantations had a better potential to restore SOC content than did cropland and grazing land [5,22], and [62] reported that the conversion of cropland to Eucalyptus plantations ameliorates soil degradation in central Ethiopia. Moreover, Assefa

et al. [63] reported that the amount of fine root biomass in Eucalyptus plantations was higher than that of cropland and grazing land.

In the lower soil depths (15–30 and 30–50 cm) at Guder, SOC and TN contents were similar to those of the surface layer, probably largely due to plant roots and exudates, dissolved organic matter, bioturbation, and translocation of particulate organic matter [64]. This result is in line with the finding of [62], who reported a similar trend across soil depths. At Dibatie, soil depth generally had no effect on SOC content, however in the lower depths TN content was higher in bushland than in grazing land. This may be because of regular burning of the surface cover in woodland, which is the most common soil fertility problem in the lowlands of northwestern Ethiopia [65], as well as leaching and lower temperature in the subsurface layer [55].

Land use had a significant effect on both SOC and TN stocks across the entire 50-cm soil profile at Guder (Table S1). The topsoil layer of bushland stored significantly greater SOC and TN stocks than that of plantation, which may be largely due to less carbon input from litter biomass, roots, and residues, including understory biomass in plantations [10,12,66,67]. At the lower two depths, however, bushland and grazing land had the highest SOC and TN stocks. Similarly, at Aba Gerima, SOC and TN stocks of cropland were lower than those of the tree- and grass-based systems of bushland, plantation, and grazing land. Many studies have reported that cropland stores the lowest SOC and TN stocks [27,52]. In the 30–50 cm soil layer, bushland also showed higher SOC and TN accumulation than cropland (Table S1). At Dibatie, however, SOC and TN stocks were similar at all soil depths, except for the TN stock in the lower soil depth. This could be due to the practice of burning woodland (as discussed above).

### 4.3. Effect of Agro-Ecosystem on SOC and TN Contents and Stocks

Agro-ecology had a significant effect on SOC and TN contents and stocks (Table 4). The soil under bushland and grazing land had lower SOC and TN contents at Aba Gerima and Dibatie than at Guder. In different ecosystems, climate strongly affects the soil carbon and nitrogen by controlling vegetation productivity and organic matter decomposition [68]. Guder had higher mean annual precipitation and was cooler than the other two sites (Figure 2). Similar studies also reported that areas with high mean annual precipitation and lower mean annual temperature tend to accumulate large amounts of SOC and TN [5,49,68]. The SOC and TN contents in cropland were lower at Aba Gerima than at Guder and Dibatie. This result clearly indicated that cropland at Aba Gerima had less organic input and poor physical protection, including vegetation cover, which plays a substantial role in organic matter stabilization in cultivated land [69]. In another study of agro-ecosystems of the Upper Blue Nile basin, Ebabu et al. [43] reported greater soil loss for cropland at Aba Gerima than that at Guder and Dibatie. However, plantations had similar SOC and TN contents at Guder and Aba Gerima.

The SOC stock in bushland (141.19 Mg ha$^{-1}$) and grazing land (109.94 Mg ha$^{-1}$) was greater at Guder than at Aba Gerima. These values are comparable with previous reports of SOC stocks of 69–239 and 67–109 Mg ha$^{-1}$ in natural vegetation and grazing land to 50 cm depth in the northwest highlands of Ethiopia [5,62], but they are markedly higher than the values reported by [12], who recorded SOC stocks of 52 and 39 Mg ha$^{-1}$ to 50 cm depth in grazing and shrub land of northern Ethiopia, respectively. These values are also lower than the estimated mean of tropical sites (216 Mg ha$^{-1}$; [35]) and the global average (254 Mg ha$^{-1}$; [70]). However, cropland had higher SOC and TN stocks at Dibatie than those at Guder and Aba Gerima, which could be related to the different farming system at Dibatie. Unlike at Guder and Aba Gerima, crop residues are not collected in the field at Dibatie, which could be contributing to the SOC accumulation. In addition, the bulk density in cropland at Dibatie was higher than that at the other sites (Table 3). On the other hand, cropland at Dibatie is a new land-use type, having been converted from woodlands (Combretum–Terminalia) recently. In southern Ethiopia, [71] reported that soil under Combretum–Terminalia vegetation stored higher carbon stock than the aboveground biomass.

*4.4. Implications of SOC and TN Stocks as Indicators for Sustainable Land Management in the Upper Blue Nile Watershed*

At the watershed scale, the effects of topographic position and land use on SOC and TN stocks were not consistent. At Guder, Aba Gerima, and Dibatie, topographic position and land use, land use, and topographic position, respectively, were the dominant factors that affected SOC stock (Table 2). However, TN stock in all agro-ecosystems was affected by topographic position and land use. Thus, by maintaining the same land uses at Guder, both stocks of SOC and TN could be enhanced by topographic position, whereas converting bushland and grazing land to A. decurrens woodlots would likely diminish the SOC and TN stored in the soil. At Aba Gerima, conversion of cropland to Eucalyptus plantation had a positive impact on SOC and TN [5,62]. Plantation had lower SOC and TN stocks due to poor undergrowth and litter removal [72]. The interaction of land use and topographic position showed a significant effect on SOC and TN stocks at Guder and Aba Gerima (Table 2), indicating that the variation in topography and land use may simultaneously affect different soil processes including soil erosion and the accumulation and decomposition of organic matter [10,30].

Across the agro-ecosystems, topographic position and land use were the main factors influencing SOC and TN stocks, but agro-ecosystem also showed a significant interactive effect with topographic position and land use on the SOC and TN stocks (Table 4). Among agro-ecosystems, SOC and TN stocks were higher at Guder, followed by Dibatie and Aba Gerima (Figure 4). In addition to vegetation composition, the hydrological regime, soil formation processes, and climate (temperature and precipitation) are important factors that affect the SOC [68], which in turn influences soil respiration [33]. In this study, Guder has higher mean annual precipitation and lower mean annual temperature (Figure 2a) than Aba Gerima (Figure 2b) and Dibatie (Figure 2c). Agro-ecosystems in cooler and moister climates accumulate high SOC and have a low rate of soil respiration [33] and limited microbial activity [50]. A warm and moist agro-ecosystem such as Dibatie, however, tends to store moderate SOC stocks due to high biomass production (Table 1) and greater soil respiration. Aba Gerima has low SOC and TN stocks, likely as a result of severe soil erosion, prolonged crop cultivation, and poor land management. The C/N ratio varied from 11–13 at Guder to 10–21 at Aba Gerima and 15–18 at Dibatie. The C/N ratio is commonly considered as an indicator of microbial activity and quality of soil organic matter [73]. Similar to the SOC and TN stocks, the C/N ratio also varied among land-use types, agro-ecosystems, and topographic positions.

## 5. Conclusions

This study clearly demonstrated that SOC and TN stocks varied significantly across land-use types and topographic positions of different agro-ecosystems. Poor and environmentally damaging land management practices tended to reduce SOC and TN in soil. Interactive effects of topographic position and land-use types on SOC and TN stocks were significant at Guder and Aba Gerima. Bushland at Guder accumulated a substantial amount of SOC and TN stocks. Cropland at Aba Gerima had poor SOC and TN stocks. Compared to other land-use types, the soil of *A. decurrens* plantation was the lowest in SOC and TN, due to high biomass removal and improper silvicultural management. However, *E. camaldulensis* plantations at Aba Gerima had a positive impact on SOC and TN stocks. Across agro-ecosystems, Guder and Dibatie accumulated larger SOC and TN stocks than those of Aba Gerima.

Overall, land use was a crucial factor influencing SOC and TN, both within and across the sites. However, the effect of topographic position was more pronounced across watersheds than within them. Aba Gerima showed lower SOC and TN stocks due to prolonged crop cultivation and mismanagement of the landscape. This calls for immediate land management interventions, particularly targeting croplands. Our findings highlight the importance of assessing SOC and TN stocks when designing evidence-based land management options in the Upper Blue Nile basin.

**Supplementary Materials:** The following are available online at http://www.mdpi.com/2071-1050/12/6/2425/s1. Table S1: SOC and TN stocks at three soil depths under different land-use types at Guder, Aba Gerima, and Dibatie. Table S2: SOC and TN contents and stocks at the three topographic positions of the study sites.

**Author Contributions:** Conceptualization, G.A. and N.H.; data curation, G.A. and K.E.; validation, A.T. (Atsushi Tsunekawa) and N.H.; methodology, G.A., M.W., T.M., and T.T.; formal analysis, G.A.; investigation, G.A.; resources A.T. (Atsushi Tsunekawa) and N.H.; writing—original draft preparation, G.A.; writing—review and editing, A.T. (Atsushi Tsunekawa), E.A., N.H., T.T., M.W., K.E., and M.L.B.; supervision, A.T. (Atsushi Tsunekawa), N.H., T.T., and M.T.; project administration, A.T. (Atsushi Tsunekawa), E.A., N.H., and A.T. (Asaminew Tassew); funding acquisition, A.T. (Atsushi Tsunekawa), E.A., and N.H. All authors have read and agreed to the published version of the manuscript.

**Funding:** This research was funded by the Science and Technology Research Partnership for Sustainable Development (grant no. JPMJSA1601), Japan Science and Technology Agency/Japan International Cooperation Agency.

**Acknowledgments:** We are grateful to Anteneh Wubet and Agerselam Gualie for the facilitation of our field and laboratory work. We also thank the Arid Land Research Center of Tottori University for providing a convenient research environment and facilities throughout our work.

**Conflicts of Interest:** The authors declare no conflict of interest.

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
