# Peer review of "Effects of Land Use and Topographic Position on Soil Organic Carbon and Total Nitrogen Stocks in Different Agro-Ecosystems of the Upper Blue Nile Basin"

_sustainability, doi:10.3390/su12062425_

Round 1

Reviewer 1 Report

This manuscript examined SOC and TN stocks in response to different land-use types and topographic positions in agroecosystems, providing insights on soil management for sustainable agricultural practices. From this perspective, this manuscript falls into the scope of this journal. Overall, this manuscript is well-organized and the experiment design is scientifically valid. Here, I’ve provided some comments for the authors to consider.

I have two major comments. Firstly, the authors may put more effort into formatting their Results section. Currently, this section is too lengthy and less-organized. I would suggest the authors reduce this section by describing trends with some key values, but not all of them.

Secondly, from my perspective, I would put some doubts on results from maps of SOC and TN, cause the spatial interpolation is just too simple. There are two reasons, (1) you don’t have enough sampling points to account for the spatial structure of SOC and TN across each watershed; (2) such interpolation is just another way of presenting results as you’ve shown in Figure 5, no new information is provided.

Line comments

Line 32: here, you’ve mentioned that both SOC and TN stock varied significantly across the land uses, topographic positions, and agroecosystems. However, you have not explained what kinds of agroecosystems you have in this manuscript. It’s just confusing. Also, change stock to stocks.

Line 38, change ratio to ratios.

Line 39-41: again, it’s confusing when you were using agroecosystems but did not explain what does this term refer to.

Line 44: add on before croplands, targeting on croplands.

Lines 94-99: Again, more information about these agroecosystems would be helpful for understanding your aims. Based on abstract and study sites, it seems that these three agroecosystems refer to three locations. What kinds of aspects make these three agroecosystems different?

Figure 1: firstly, I am confused about the red polygon. In the figure caption, you’ve mentioned that red polygon refers to topographic position, but you have three topographic positions. Secondly, I would suggest the authors add a scale bar for the upper Blue Nile basin.

Lines 113-116: you stated that Guder had the highest mean annual rainfall (2454mm), however, in the figure, it seems that Aba Gerima (b) had the highest mean annual rainfall, please clarify.

Line 118: Please change Figure 1 to Figure 2. Also, I would suggest no abbreviation for figure legend.

 Line 136: I would suggest the authors provide more information on topographic positions for each watershed, for example, mean elevation, slope, etc. Also, the authors referred to these three watersheds as different agroecosystems. It’s hard to follow when you were using different terms, please be consistent.

Line 148: what are units for the last two rows (i.e., cropland and grazing land)?

Lines 160-164: I would suggest the authors change the unit for SOC and TN concentrations to mg g-1 since, in the rest of the paper, you were using this unit for concentration.

Lines 166-174: Please see my above comments for this mapping method.

Lines 180-182: Please be more specific, which analyses you used three-way ANOVA, and which you used nested three-way ANOVA.

Lines 185-186: Please specify which software you used for spatial interpolation.

Lines 201-205: How did you compute the concentrations of SOC and TN across three soil depths?

Lines 206-209: I would suggest the authors keep consistent with the previous two paragraphs, using actual values instead of folders (or times).

Lines 223-228: How did SOC stock change with topographic positions?

Lines 248-299: This is too much, please consider being more concise. From Figure 6, two general trends were observed, (1) both SOC and TN concentrations decreased slightly with soil depths, (2) the impacts of land use on SOC and TN concentrations persisted to 50 cm of the soil profile. 

Lines 308-310: What the main difference between table 4 and Figures 4 and 5? This table seems to be redundant.

Lines 300-346: Some of these results were just simply repeated from previous results, please be concise.

Lines 391-409: It comes to me that some of these land-use types are closed related to agricultural activities. Do you think that the application of fertilizer will interact with other factors (like topographic position) to affect soil TN concentration and stock? This might be worth discussing.

Lines 466-492: Again, what characteristics (except highland, midland, lowland) make these three agroecosystems different? I am still not quite sure how you would like to define the agroecosystem.

Lines 495-524: I am glad you have this discussion which will be helpful for developing soil management practices for this region. 

Author Response

Manuscript ID: sustainability-715799

Title: Effects of land use and topographic position on soil organic carbon and total nitrogen stocks in different agro-ecosystems of the Upper Blue Nile basin

We are very grateful for the reviewer for the important comments, which help us to improve our manuscript thoroughly!

Please find below a discussion of the main issues raised by the reviewer and the way these have been taken into account in the revised version of the manuscript. Please note that following each comment the corresponding discussions are given under sub-heading ‘Response’ in blue-colored fonts. Line numbers referred in response are related to the revised manuscript. Where in the changes we have made in the revised manuscript are shown by track changes.

Sincerely,

Getu Abebe

Reviewer 2 Report

A well designed and executed research with very good findings which confirmed experimentally theoritical aspects. A few observations/suggestions are marked on the text.

Author Response

Response to the Reviewer Comments

Title: Effects of land use and topographic position on soil organic carbon and total nitrogen stocks in different agro-ecosystems of the Upper Blue Nile basin

We are very grateful for the reviewer for the important comments which help us to improve our manuscript thoroughly!

Please find below a discussion of the main issues raised by the reviewer and the way these have been taken into account in the revised version of the manuscript. Please note that following each comment the corresponding discussions are given under sub-heading ‘Response’ in green-colored fonts. Line numbers referred in response are related to the revised manuscript. Where in the changes we have made in the revised manuscript are shown by track changes.

Sincerely,

Getu Abebe

Round 2

Reviewer 1 Report

The authors have addressed my comments, congrats on a nice work!